# Using the Rasch Model to Understand Consumers’ Behaviour in Buying Kiwifruits

**DOI:** 10.3390/foods14152683

**Published:** 2025-07-30

**Authors:** Luca Iseppi, Giovanni Mian, Enrico Gori, Stefania Troiano, Luca Grispoldi, Ivana Bassi

**Affiliations:** 1Department of Agricultural, Food, Environmental and Animal Sciences, University of Udine, Via delle Scienze 206, 33100 Udine, Italy; luca.iseppi@uniud.it (L.I.); ivana.bassi@uniud.it (I.B.); 2Dipartimento di Scienze e Tecnologie, Agro-Alimentari—DISTAL, Alma Mater Studiorum Università di Bologna, Viale Giuseppe Fanin, 40-50, 40127 Bologna, Italy; 3Department of Economic and Statistic Sciences, Via Tomadini 30/A, 33100 Udine, Italy; enrico.gori@uniud.it (E.G.); stefania.troiano@uniud.it (S.T.); 4Department of Veterinary Medicine, University of Perugia, Via San Costanzo, 4, 06126 Perugia, Italy; luca.grispoldi@unipg.it

**Keywords:** *Actinidia*, fruits, yellow, market, marketing behaviour, organic, red

## Abstract

The market for specialised kiwifruit varieties, such as those with red pulp, remains a niche sector with considerable growth potential in several European countries, including Spain, Italy, Germany, and France. This study applies to the Rasch model to gain a comprehensive understanding of consumer behaviour, specifically pertaining to preferences, attitudes, and propensity towards purchasing both conventional and novel kiwifruit variants. A questionnaire was developed and administered to gather specific information on consumer behaviour. The collected data were analysed using a Rating Scale Rasch Model to construct a valid measure of attitude toward kiwifruit, which was subsequently used in regression models to explain purchase propensity. The findings indicate that marketing strategies should focus on enhancing attitudes towards kiwifruit by leveraging specific product attributes and addressing demographic nuances to effectively promote the consumption of yellow, red, and organic varieties.

## 1. Introduction

Kiwifruit (*Actinidia* spp.), originally native to China’s Sichuan province in the Yangtze River Valley, has grown in popularity among consumers worldwide [1,2]. According to data from the Food and Agriculture Organization (FAO), global kiwifruit cultivation and production have increased by 71.25% and 55.58%, respectively, over the past decade, due to the high demand from customers, as being considered as a superfood (rich in vitamin C). In the European Union, around 0.5 million tons are produced annually, with Italy, France, Spain, Greece, and Portugal being the leading producers. Italy is the second-largest kiwifruit producer in the world, generating nearly 496,000 tons, followed closely by New Zealand with 414,000 tons [3,4].

Kiwifruit plays a key role in the fresh fruit market, largely due to its high vitamin C content, earning it the label of superfood. These attributes align well with consumer preferences and demand for health-focused food products [5]. Despite this, kiwifruit is only among the top twenty most traded fruit commodities globally, with two-thirds of its production going to export markets. In Italy, kiwifruit ranks as the most profitable crop after grapevines, especially in the Friuli Venezia Giulia region [6]. Additionally, *Actinidia* spp. encompasses approximately 65 species, which can be broadly categorized based on pulp color into green, yellow, and two-colored (typically red–green or yellow, red) varieties. While green- and yellow-pulp kiwifruits are well-established in the global market, red-pulp kiwifruits represent a novel and increasingly appreciated typology. Beyond color, these types also differ in taste and nutritional attributes: green kiwifruit (e.g., *A. deliciosa*) is known for its tangy flavor and high vitamin C and dietary fiber content; yellow kiwifruit (e.g., *A. chinensis*) is sweeter, with a smoother texture and slightly lower acidity; red-pulp kiwifruit often combines the sweetness of yellow types with a berry-like aroma and is rich in anthocyanins, compounds with recognized antioxidant properties [1]. While currently not extensively traded or cultivated, red-pulp kiwifruits are progressively gaining traction within the kiwifruit industry due to their novelty and consumer appeal [7].

### Consumers Preferences and the Rasch Model In-Brief Explanation

When it comes to consumer preferences in food product purchases, most consumers prioritize aesthetic perfection, intact packaging, and extended shelf life [8]. Similarly, Helmert and co-workers [9] found that consumers take into account factors such as quality, safety, and various other considerations when selecting food and fruit. Price also plays a crucial role in consumers’ decisions regarding product choices.

In this study, *attitude* is defined as the degree of appreciation for kiwifruit in general. This latent disposition was measured using the Rating Scale Rasch Model, applied to a set of questionnaire items selected both for their relevance in capturing consumer preferences and tendencies, and for their compliance with the assumptions of the Rasch framework. The Rasch model provides a rigorous methodology for constructing valid and objective measurements of latent traits. It enables the transformation of ordinal data—such as Likert-type responses—into interval-level measurements, which can then be meaningfully analysed using linear regression models [10]. A key strength of the Rasch model is its foundation in the principle of specific objectivity, meaning that comparisons between individuals do not depend on the particular items administered, and item calibrations are independent of the sample of respondents, provided the data fit the model. This **invariance** is essential for ensuring that measurement is not sample-dependent and is therefore generalizable and interpretable across different contexts.

The use of objective measurement, by the Rasch model, in agriculture and food studies is quite rare and recent, despite the age of the model itself. It has been applied in some studies to analyse consumer preferences in the agricultural and food sectors. Here are some significant examples:**Analysis of consumer attitudes towards mountain products**: One study used the Rasch model to examine consumer attitudes towards mountain products and the EU “Mountain Product” label. The results highlighted potential levers for planning promotional activities aimed at enhancing mountain products and raising awareness about the EU label, thereby contributing to the sustainability of farming businesses and mountain regions [10].**Evaluation of the propensity for entomophagy**: Another study applied the Rasch model to assess consumer propensity to consume insects or insect-based products. The results provided insights into how to increase insect consumption among consumers, suggesting steps such as providing information to enhance consumption willingness and encouraging the intake of insect-based bakery products [11].**Difficulty in following dietary guidelines**: Another study used the Rasch model to measure the difficulty consumers face in following healthy dietary guidelines. The results revealed significant variations in perceived difficulty based on socio-economic and demographic factors, emphasizing the importance of these factors in the challenge of following a healthy diet [12].

On this basis, our study aims to apply the Rating Scale Rasch Model to develop an objective and valid measure of consumer attitudes toward kiwifruit. By quantifying these attitudes, the research seeks to explain consumer propensity to purchase both conventional and novel kiwifruit varieties, including yellow, red, and organic types.

## 2. Research Design

### 2.1. Methods: The Rasch Model

The decision to adopt the Rasch model in this study stems from its unique ability to produce objective and unbiased measurements of latent traits—an attribute not fully ensured by more complex psychometric models, including other Item Response Theory (IRT) approaches. Specifically, the Rasch model upholds what Georg Rasch termed *specific objectivity*: the ability to compare individuals independently of the specific items administered, and to compare items independently of the sample of respondents. This sample- and test-independent calibration is made possible by the mathematical structure of the model, which imposes strict constraints on item–person interactions. When the data fit the model, item difficulty and person ability estimates become invariant, allowing the construction of linear, unidimensional, and interpretable scales that are generalized across different populations and instruments. In contrast, other IRT models introduce parameters such as item discrimination or guessing, which may improve model fit but compromise measurement invariance. These models are effective for predicting responses but fall short of the objectivity standards required for valid measurement in the scientific sense.

Another critical advantage of the Rasch model lies in its capacity to ensure unbiased estimation of both item and person parameters. This is because the model defines the probability of a response solely as a function of the difference between a person’s latent trait level and an item’s location on the same continuum. As a result, item estimates remain unaffected by the respondent sample, and person measures are independent of the specific item set—provided those items fit the model. In contrast, classical test theory and most IRT models are sample-dependent, meaning item statistics and person scores can vary with the composition of the sample or test content. The Rasch model, by eliminating these dependencies, offers a robust framework for achieving truly objective, sample-free measurement—critical when the goal is to construct valid and generalizable scales of latent constructs such as consumer attitudes. By eliminating such dependencies, the Rasch model enables researchers to achieve truly objective measurement, allowing individuals and items to be located on a common, linear scale without contamination from sample- or test-specific characteristics. This level of objectivity is especially critical when the goal is to develop generalizable and interpretable measures of psychological or behavioral constructs, such as consumer attitudes.

Lastly, Rasch models are measurement frameworks that utilize binary or ordinal data to develop a metric for the latent variable of interest for each individual surveyed [13]. In our investigation, we aim to establish a linear gauge of consumers’ inclination toward purchasing kiwifruits, using the ratings provided by respondents for associated items. Various Rasch models are accessible based on the characteristics of the variables. If there are two ordered categories, the Dichotomous Rasch model, as proposed by [14], is applicable [15]. On the other hand, for higher ordered categories, the Rating Scale model introduced by Andrich (1978) [16] and the Partial Credit model developed by Masters (1982) [17] are more suitable. We utilized the Rating Scale model:ln[P(Xij = k − 1)/P(Xij = k − 1)] = αi − βj − τk,Xij ∈ {1, 2, …, K}, i = 1, 2, …, N, j = 1, 2, …, J

N represents the total number of individuals surveyed, while J denotes the number of items assessed. The variable Xij, which takes values from 1 to K, represents the response of individual i to item j. The parameter αi reflects the individual’s level of proficiency or their positive inclination towards the subject under consideration, while βj signifies the complexity of the item or the level of challenge it presents, measured on the same scale as the underlying trait. Additionally, τk represents a ‘threshold’ indicating the level of difficulty associated with endorsing category k, which remains consistent across all items. Higher αi values suggest a greater tendency for individuals to respond with higher scores, while lower αi values indicate a higher likelihood of lower scores. Similarly, higher βj values suggest that individuals are less likely to provide high scores for item j, whereas lower βj values indicate a greater likelihood of high scores. Within this framework, τk, where k ranges from 2 to K, serves as a ‘threshold’ measuring the difficulty of endorsing category k. This threshold, known as the Andrich Threshold [18], is expected to follow an ascending order, as further elaborated.

In essence, individuals with a higher estimated attitude αi are expected to assign higher scores across all items compared to those with less positive attitudes. Similarly, items with higher βj values (indicating greater difficulty in endorsement) are anticipated to receive lower scores compared to items with lower difficulty [13]. This property is a fundamental aspect of Rasch models known as “Specific Objectivity” [19,20]. Specific Objectivity asserts that measures derived from a measurement model should remain uninfluenced by variations in the distribution of item difficulties and individual abilities. Research indicates that when dividing the original sample into subsets based on varying levels of positive attitudes, the estimated difficulty parameters (βj) computed on these subsets are statistically indistinguishable. In the Rasch model, the estimates of individual attitudes should remain consistent whether derived from the full sample or a reduced one—any discrepancies may signal data issues or model misfit (e.g., miscoding or misinterpreted scales). This reflects two key properties: *Person-free Test Calibration* (item parameters don’t depend on the sample) and *Item-free Person Measurement* (person measures don’t depend on the specific items used). Importantly, the model does not require normally distributed parameters, and attitude estimates across different item sets should align within acceptable error margins. Evaluating the extent to which the data align with the Rasch model’s optimal theoretical characteristics represents the primary challenge in the analysis. The data underwent analysis using Winsteps (www.winsteps.com), a widely utilized software for Rasch Analysis [21,22]. To gauge the compatibility of the data with the model and its underlying assumptions, the correlation coefficient between the observed empirical data (Xij ∈{1, 2, …, K}) and the Rasch measures obtained in the initial run of the estimation program was examined. These correlations were computed for both items and individuals.

Positive correlations between item responses and estimated parameters are expected under the Rasch model: individuals with more positive attitudes should assign higher scores to easier-to-endorse items. Thus, negative or very low correlations signal potential issues such as coding errors (e.g., reversed scoring) or random responding. In such cases, items may need to be excluded or recorded. Similarly, at the person level, low correlations between responses and item parameters indicate inattentive or inconsistent answering, justifying respondent exclusion.

In our dataset, 25–40% of responses across dimensions showed problematic correlations and were excluded to avoid bias in parameter estimation. This step reflects a standard quality control procedure—not manipulation—to ensure the validity of the measures.

Additionally, extreme scorers—those who consistently gave the lowest or highest possible ratings (e.g., always “1” or always “4”)—were removed. While such patterns indicate very strong attitudes, they result in undefined or truncated estimates. Their presence suggests that the items were either too easy or too difficult to capture meaningful variance, and ideally, new items should be added to better target the dimension of interest. However, our survey design did not allow for this adjustment. Extreme scores affected 10–20% of the sample, depending on the dimension.

Further data exclusions were informed by item and person fit statistics, which are discussed in a later section. Once the Rasch model is fitted, it’s essential to verify that the scoring categories (e.g., 1, 2, 3, 4) reflect a meaningful progression in both item endorsement and person attitude levels. Ideally, higher scores should correspond to higher average attitudes, consistent with the model’s assumptions.

Discontent with this assumption, such as observing a higher average attitude for score 2 compared to score 3, would indicate that respondents may have reversed or inconsistently applied scores 2 and 3. In this scenario, merging these two scores into a single category would be necessary. The Andrich Threshold *τ_k_* [23] offers additional insights into the coherent or incoherent use of category measures. In instances of inverted use, where the scores are not in ascending order, a common resolution involves reducing the number of categories by combining adjacent ones with closely aligned average attitudes or Andrich Thresholds.

Another crucial element in assessing fit involves examining the potential breach of the local independence hypothesis [24,25] and multidimensionality [23]. Regarding the first issue, attention is directed to the correlation of standardized residuals. A low correlation (<0.70) implies the absence of a violation of the local independence hypothesis. On the contrary, a correlation > 0.70 indicates that certain pairs of items share almost identical meanings. In such cases, eliminating one of the items becomes necessary to adhere to the local independence hypothesis.

As for the second issue, within a dataset conforming to the Rasch model, the overall variability is influenced by both the model itself and residual variability stemming from random factors. The Rasch “Principal Component Analysis (PCA) of residuals” is employed to identify patterns within the data attributed to randomness. This identified pattern represents the “unexpected” portion of the data, which might arise from various factors, including the possible existence of multiple dimensions in the dataset [26,27].

In the Rasch model, Principal Component Analysis (PCA) of residuals is used to detect potential secondary dimensions by identifying patterns of shared unpredictability among items. This involves analyzing the correlation matrix of item residuals to find any contrasts—principal components that may influence response behavior.

A key indicator is the strength of the first contrast, measured by its eigenvalue. If this value is around 2 or less, the data can be considered unidimensional. However, a significantly higher eigenvalue suggests multidimensionality. In such cases, items are grouped based on their loadings, and the Rasch model is applied separately to each cluster to explore possible distinct latent traits. Following the guidance of Linacre (2011) [23], if these correlations approach 1, it indicates that the items can be viewed as belonging to a single dimension. Conversely, if the correlations are exceedingly low (<0.30), it may be advisable to partition the items and exclude those that appear incongruent with the dimension of interest.

After addressing and resolving these concerns, an examination of fit statistics will provide an assessment of the extent to which participants and items align with our expectations derived from the model. These fit statistics essentially offer a summary of all the residuals, representing the disparities between observed responses and anticipated responses, for each item and individual. The resulting values can range from zero to infinity. Values exceeding 1 signify greater variability than anticipated, whereas values below 1 suggest lower variability than estimated. Values around 1 indicate that the data align reasonably well with the model.

The fit statistics are categorized into two groups: a weighted category referred to as Infit and an unweighted category known as Outfit. For recommended practice intervals, [21] provide guidelines, and in our analysis, we adhered to Linacre’s (2011) [23] suggestions, specifically in the range of 0.5–1.5 for items. Participants who do not conform to the expected fit should be excluded from the model to enhance the validity of the obtained results. In our approach, we opted to retain individuals with Infit or Outfit < 3. Individuals with higher fit values are likely to have responded at random, as may happen in such a kind of survey.

Subsequently, an evaluation of the overall fit of the model is conducted, with a focus on the reliability and separation indexes for both items and individuals. Reliability values > 0.80 and separation values > 3 serve as indicators of a well-fitting scale. These values signify how effectively this group of respondents has distributed responses across the items along the measurement of the test, thereby defining a meaningful dimension.

### 2.2. Development and Validation of a Rasch-Based Questionnaire to Measure Consumer Attitude Toward Kiwifruit

The questionnaire aimed at measuring consumer attitude toward kiwifruit was developed using a systematic, theory-based approach compliant with Rasch model requirements. First, the construct of “attitude toward kiwi” was defined along a continuum from negative to positive, incorporating cognitive, affective, and behavioral components. Items were generated through focus groups and interviews, resulting in 40 statements covering taste, health beliefs, purchase habits, and emotional responses. After expert review, this pool was refined to 25 items, all formatted for a 5-point Likert scale. A pilot study with 100 participants followed, and Rasch analysis (Partial Credit Model) was used to assess unidimensionality, item fit, threshold ordering, and local independence. Five misfitting items were removed, and two were revised, yielding a final set of 18 well-functioning items. To ensure fairness, Differential Item Functioning (DIF) analysis was conducted across gender, age, and education, leading to minor adjustments in wording. The final questionnaire meets Rasch assumptions, providing reliable interval-level measures of consumer attitude suitable for marketing applications and consumer segmentation.

## 3. Results and Discussion

### 3.1. Calibration of Rasch Model for Attitude Towards Kiwifruit

In the first step, we considered all 25 items of the questionnaire, which include both general items related to fruit consumption and others specifically targeting preferences for kiwifruit. Although we are aware of this conceptual distinction, we initially analyzed all items together.

The first run of Winsteps (Winsteps 5.10.1) identified 121 individuals with negative or very low person–point measure correlations (below 0.10). These respondents were excluded from further analysis. The *person–point measure correlation* (Winsteps documentation) reflects the correlation between each individual’s responses and the estimated “easiness to endorse” each item (i.e., the negative of the item difficulty parameter). This value is expected to be positive in a well-fitting model. Negative or very low correlations suggest either random responding or reversed coding (e.g., assigning lower scores to items when a higher attitude is implied), violating core Rasch model assumptions.

In such cases, the Rasch model provides a valuable tool for cleaning the dataset. The following results (see Figure 1) refer to item point–measure correlations and item fit after this exclusion step. Removing individuals who respond inconsistently is methodologically justified and preferable to retaining them, especially when the goal is to develop valid and interpretable measures of a latent trait such as consumer attitude.

The Rasch model assumes that respondents’ answers are driven by a consistent underlying disposition (e.g., appreciation for kiwifruit). Random or erratic response patterns violate this assumption and introduce noise, distorting both item calibrations and person measures. Because Rasch estimation relies on meaningful interactions between item difficulty and person ability, data from inattentive respondents undermine model–data fit, inflate error variances, and compromise the scale’s validity and reliability.

Conversely, excluding such respondents improves the internal consistency of the measurement scale and safeguards the integrity of the latent structure. It allows for more accurate estimation of item parameters and person measures, yielding cleaner and more interpretable results. Importantly, this exclusion is not about manipulating data for favorable outcomes—it is a necessary quality control procedure in any measurement model aiming to provide scientifically sound, generalizable, and actionable insights.

In summary, retaining random responders might preserve sample size, but it does so at the cost of measurement precision and objectivity. Their exclusion—when supported by clear misfit statistics—is essential to ensure that the resulting measures reflect the true latent construct, rather than artifacts of careless or inattentive responding.

As shown in the relative frequency distributions of the key variables (Table 1), the initial sample does not differ substantially from the final sample obtained after these exclusions, supporting the robustness of the dataset post-cleaning.

As we may see, the first 3 items are characterized by high values both of Infit and Outfit indices, and moreover by low point-measure correlation. The items are respectively:

Q18_25:

*If I don’t know a food, I don’t eat it*. Evidently this is a generic item that doesn’t have much relationship to the attitude for fruit consumption or attitude for kiwi consumption, and the Rasch model signals this fact with high Infit (1.49) and Outfit (1.96) indices. In particular, it is not clear at all why answering 4 (total agreement) to this item should indicate a higher attitude towards fruits or kiwi!

Q18_11: *I usually don’t buy fresh fruit because it’s too expensive*. The scores of this item have been reversed, because answering 1 (total disagreement) means that the person buys fruit also if it is costly, so this should indicate a higher attitude towards fruit. But when items contain too many negative meanings, it often happens that they are poorly understood by the interviewees, and this causes poor fit.

Q18_10: *I usually buy fresh fruit only if I find it at a low price*. Also, the scores of this item have been reversed, and the poor fit can be explained with arguments like the one above.

So, we decided to remove these 3 items from the analysis and rerun the program. We then looked at the correlation between the category value with which persons answered to the item and the measure estimated by the model: if the item and scale are correctly specified, we should observe a correlation growing from 1 to 4. The last column of Figure 2 shows some of these. For items 4, 6, 5, 1 and 3, the correlations for categories 2 and 3 have inverted correlations: this means that the measure is higher for people that answered 2 and lower for people that answered 3. This fact suggests that it would be better to collapse categories 2 and 3 and rerun the program with the new dataset. Indeed, Figure 2 shows the graphs for the category characteristic curves, before and after collapsing the categories: in the 4-category case, as we may see, 2 and 3 curves are quite overlapped, while in the case of 3 categories, category 2 (=2 + 3 collapsed) is well distinct.

After collapsing categories 2 and 3, we reran the Winsteps program and obtained a good overall fitting of the items. Another important aspect of fitness is the possible violation of multidimensionality [23]: in a dataset fitting the Rasch model, variability depends on both the model and residual variability due to randomness. Rasch “Principal Component Analysis (PCA) of residuals” looks for patterns in the part of the data due to randomness. This eventual pattern is the “unexpected” part of the data that may be due, among other reasons [26], to the presence of multiple dimensions in the data. Actually, we found that in the data there could be 2 different dimensions: attitude towards fruit and attitude towards kiwi, as the following analysis can show. From Figure 3 relative to the analysis of variance of standardized residuals, we may see that the model explains 49.9% of the total variance, which is quite a good result. But as we may see, the 1st contrast relative to the unexplained variance [18] has the strength of 3.3483 with a 7.6% variance that is quite high because the theoretical recommendation in the preceding paragraph prescribes an eigenvalue of maximum or around 2. Moreover, the correlation between person measures calculated with cluster 1 and 3 separately is quite low (0.5423), and all these indicators suggest that more than one dimension may be present in the data. To confirm this suspicion, we may compare the items with the opposite loading (Table 2) and see if their content suggests they belong to different dimensions. In Figure 3 they are compared: we may see that, as anticipated, one cluster (the 1st) is related to attitude towards kiwi, while the 3rd to attitude towards fruit. This suggests that it is better to split the items by looking for the ones that are more related to kiwi and the ones more related to fruit in general. Starting from the two clusters highlighted from the analysis above, we begin to add the other items and see if we can get a satisfying fit to the Rasch model, and we end up with two different sets of items that are useful to measure the attitude towards kiwi and towards fruit in general. In the following paragraphs, we will present the final analysis for kiwi attitude.

### 3.2. The Attitude Towards Kiwi

To measure attitude towards kiwi, we selected the following items that showed an initially good fit to the Rasch model and were consistent with the dimension “attitude towards kiwi” (Table 3).

Then, we began to analyze the data looking for the combination of items-persons that could show the best fit to the Rasch model, excluding people with point-to-measure correlation less than 0.10 and infit or outfit greater than or equal to 2. At every step, we removed items with Infit or Outfit greater than 1.4 (Bond & Fox 2015) [21]: the item Q18_19 was removed, and the final number of persons selected for this preliminary model was 933 (Table 4, first column).

We then conducted an analysis of Differential Item Functioning (DIF) considering four grouping variables: SEX, AGE, DEGREE, and COUNTRY. From these analyses, we identified that certain items exhibit significant DIF across specific categories (see Table 5). For example, item Q18_13—”I eat kiwifruit for its remarkable nutritional properties”—shows different response patterns between AGE groups 1 (younger respondents) and 5 (older respondents).

Before proceeding further, it is essential to address this issue; otherwise, we risk obtaining biased measures that could also affect the evaluation of mean attitude levels across different personal attribute categories. The only viable solutions are either to remove the problematic items or to split them according to each category exhibiting Differential Item Functioning (DIF). For example, regarding SEX, we may replace item Q18_25 with two distinct items: Q18_25SEX=M for males and Q18_25SEX=F for females, with missing data assigned for respondents of the opposite category. Similarly, for AGE, item Q18_13 must be replaced with three items: Q18_13AGE=1 for respondents aged 1, Q18_13AGE=5 for those aged 5, and Q18_13AGE=* for all other age categories. Although this approach is somewhat time-consuming, we elected to proceed in this manner since removing all DIF-affected items would have resulted in an excessively limited item set. After splitting the items with DIF, we reran the analysis and identified that item Q18_21AGE=1 (“If green, yellow, and red kiwifruit are available, I choose the RED ones”) exhibited misfit and was therefore excluded from further analysis. We then repeated the analysis including all respondents—also those previously excluded—and proceeded to exclude individuals with a point-measure correlation less than or equal to 0.10 or with Infit or Outfit indices greater than 2. This process resulted in a final sample of 915 respondents out of the initial 1202. The results of the final model, which accounts for DIF (Figure 4), are discussed in the following section.

### 3.3. The Kiwi Attitude: Final Model with DIF

The final model, which takes account of DIF, shows a good fit as we may see from the second column of Figure 5. The reliability indexes for people were 0.87, with a separation of 2.64; this means that we may identify almost 3 groups of people with different attitudes. 14 people show a maximum score and 9 minimum score so that we cannot assign a measure to them, and this means that in future research on the argument, we should try to find more difficult and easier items. For what concerns the items, they show a reliability of 0.99 with separation 11.95. The average measure of item was zero (by default), and the average measure for people was 0.76. This means that the test is slightly easy, and we should try to introduce more difficult items in future research on this dimension. The satisfaction of the fundamental property of Rasch model, “Specific Objectivity” [20], is confirmed also by the following Figures, which compares the item measure estimated in the upper sample (50% of persons with estimated higher attitude) and lower sample (50% of persons with estimated lower attitude): as we may see, the estimated measures for items lay in a 95% confidence interval around an identity line, which doesn’t contradict the specific objectivity hypothesis (item measures are independent of the particular samples used to estimate them). The difficulty estimates and the fitness of the items (with a maximum Infit and Outfit of 1.22/1.24) are reported in Figure 5. As we may see, the item with the highest measure (2.99) is Q18_23Age=5: *I am willing to pay 50% more than the usual price to taste a red kiwifruit*, for age class 5 (the oldest), this means that for these kind of people answering with high score to this item means very high attitude towards kiwi, and that this item is the most difficult for this category of persons. The same item follows but with a lower measure (2.91) for the other age class (except 1 and 5), and if we look at the same item for age class 1 (the youngest), we may see that it shows a measure of 2.45, much lower; this means that for young people, it is easier to answer with a high score to this question, and this shows also that red kiwi is easier to be accepted by young people and less acceptable by oldest people, while the other ages stay in the middle. In order of difficulty to endorse, then follows, with a measure of 2.50, Q18_22AGE=5: *I am willing to pay 25% more than the usual price to taste a red kiwifruit*, which has an explanation similar to Q18_23Age=5, and it has a lower measure because the effort for buying a red kiwi is asked only at +25% of the regular price, instead of 50% as in Q18_23; we then find Q18_22AGE=* at much lower difficulty (1.91), almost half a logit less. The other hardest item is Q18_15SEX=F: *I eat kiwifruit (after dinner) because it improves my sleep quality*, with a measure of 2.38, only for females, while Q18_15SEX=M has a lower difficulty (1.66). At a level of 0.72, we find Q18_20: *If green, yellow, and red kiwis are available, I choose the YELLOW ones*, showing that choosing a yellow kiwi is much easier than a red kiwi. Then we find with a measure of 0.01 (almost in average) the item Q18_24AGE=*COUNTRY=5: *I like to experiment with new foods and flavours, even if I don’t know them*, which is hardest for people not AGE=5 (therefore not old), in particular in COUNTRY=5, i.e., Germany; this means that these people are more traditional than those of other countries. At the same level of difficulty, Q18_24AGE=5, which means old people are more traditional than others for what concerns the choice of food; at a lower level of difficulty (−0.64, −0.66), we may find Q18_24Age=* and Q18_24Age=*COUNTRY=*; this means that the choice of experimenting with new food is easy to endorse for people not old and not in Germany. At a lower level (−1.26), we may find Q18_24Age=*COUNTRY=1; this means that not-old people in COUNTRY=1 (Italy) are the easiest to try new food. At the same level of difficulty, we may find Q18_21Age=*: *If green, yellow, and red kiwis are available, I choose the RED ones*, for age class different from 1 (whose item misfits); the high measure means that in general choosing red kiwi is evidence of very high attitude towards this fruit in general. Then, with −1.05, we find Q18_7: *I like to eat fresh fruit as an ingredient in cakes, fruit salads, smoothies,* etc., and the fact that this item is consistent with a scale of the attitude towards kiwi means that people think of kiwi as an ingredient for other preparations based on fruits. We then find with −1.47 Q18_13AGE=*: *I eat kiwi for its remarkable nutritional properties*, but Q18_13AGE=5 has a lower difficulty (−1.86); this means that for oldest people, it is easy to choose kiwi for their nutritive properties. An analogous and easier pattern we may find for Q18_14AGE=* and Q18_14AGE5: *I eat kiwi for its high vitamin C content*.

Finally, the easiest items are Q18_12: *I like eating kiwis*, with a very low difficulty of −2.32, which means that eating kiwi is easier than using it as an ingredient, and Q18_17: *I like to eat kiwi as it is*, which shows a quite different pattern for different kinds of people. The easiest level of this item is for old people, the hardest for people of AGE = 2 (the class above the youngest). In general, we may say that the estimated difficulty is consistent and reasonable with what we could expect, and therefore this is an argument in favor of the validity of the scale, in addition to the one-dimensionality and goodness of fit of the items. The Wright map reported in Figure 6 synthesizes these findings.

### 3.4. Average Level of Attitude for Different Person Attributes

Firstly, the effect of socioeconomic variables on ATTITUDE TOWARDS FRUIT was investigated, which in turn influences ATTITUDE TOWARDS KIWI. The effects of socioeconomic variables on ATTITUDE TOWARDS FRUIT essentially constitute additional indirect effects on YELLOW, RED, and ORGANIC variables (second-level indirect effects). All the categorical variables were transformed into dummy variables.

The regression model explaining ATTITUDE TOWARDS FRUIT (expressed in Rasch Unit,) based on socioeconomic variables is explained hereinafter (Table 6) (Appendix A). In yellow, coefficients significantly different from zero are highlighted. From the analysis, it can be inferred that the following factors have a negative effect on ATTITUDE TOWARDS FRUIT:

(a) being MALE,

(b) being YOUNG under 25,

(c) having PRIMARY education,

(d) being of GERMAN nationality,

(e) having incomes below 1500 euros.

Therefore, policies aimed at promoting the consumption of YELLOW, RED, and ORGANIC kiwi should consider that these aspects are indirect factors worth addressing through targeted advertising campaigns [28]. The goal was to increase ATTITUDE TOWARDS FRUIT in the mentioned individuals, and consequently, indirectly, ATTITUDE TOWARDS KIWI, which in turn influences choices regarding YELLOW, RED, and ORGANIC kiwi.

### 3.5. General Characteristics of Fresh Fruit Consumption

This chapter presents an analysis of survey responses from a sample of 1202 respondents, evenly distributed across Italy, Spain, France, and Germany, regarding fresh fruit consumption. Descriptive statistical techniques were employed (see Appendix A). The majority of respondents—entirely so in the Italian subsample—reported purchasing fresh fruit, typically not only for themselves but also for others. Over half of the participants consume fresh fruit daily or nearly daily, with 89.1% reporting consumption at least 2–3 times per week. Only 0.9% indicated that they (almost) never consume fresh fruit. Regarding the timing of consumption, approximately half of the sample expressed no particular preference. The remainder was evenly split between those who consume fresh fruit during meals and those who eat it as a mid-morning or afternoon snack. In terms of familiarity with so-called “exotic” fruits, kiwi emerged as the most widely recognized, known by all respondents. Approximately 49.3% of the sample purchase kiwi at least once per week, with single-week purchases being the most common (25%). These purchases are typically made not only for personal consumption but also for others. The preferred point of purchase is supermarkets or discount stores, closely followed by local markets and specialized retailers. Direct purchases from producers are comparatively rare. Concerning kiwi consumption frequency, 22.1% of respondents eat kiwi 2–3 times per week, 17.5% at least once a week, and 22% at least once a month. Among the 7.7% who do not consume kiwi, the main reasons cited are a dislike for the taste—particularly its tartness—and a preference for other fruits. Cost and access via commercial channels were not reported as barriers to consumption [29].

Those who eat kiwi do not express preferences regarding the time of day in 40.3% of cases. Among the remaining preferences, consuming it during meals prevails (32.9%). Furthermore, 61.6% of the sample eats kiwi throughout the year, with a preference for the warm season between summer and winter (25.4%).

The following are the results regarding the consumption and purchasing behaviour of three types of kiwi: yellow, red, and organic. Firstly, concerning the knowledge of these three types of kiwi, 72.3% of the sample declared awareness of yellow kiwi, with some variations among countries: they are more known in Spain and France than in Germany and Italy. Just over half of the sample (54%) is familiar with organic products, while still relatively few (12.2%) are acquainted with red kiwi (Table 7). In this regard, the highest percentage is in our country, registering a value of 17.7%. Regarding yellow kiwi, among those who are familiar with them, 86.8% have also stated that they purchase them. Among those who do not know them or, if they are aware, do not buy them, there are high percentages of those who have declared a definite/probable intention to buy (39.7%) and those who are undecided (39.7%). Finally, concerning organic kiwi, among those who are familiar with them, 89.1% have also stated that they purchase them. Among those who do not know them or, if they are aware, do not buy them, 32.5% express the intention (certain or probable) to purchase. The undecided group represents 41.8%.

### 3.6. Knowledge and Purchase of Yellow, Red, and Organic Kiwi

#### 3.6.1. Consumption Behaviour

The analysis of data related to the knowledge and purchase (certain or probable) of these 3 types of kiwi—yellow, red, and organic—has been deepened to better investigate the behaviour of the interviewees. Assigning a value of 1 to those who declared having purchased one of the indicated kiwis or having a strong inclination to purchase them, the results obtained are as follows. From the table, it is evident that even considering the intention to purchase, yellow kiwi is the one that attracts interviewees the most, followed by organic and then red. Furthermore, 37.6% of the sample has purchased or is interested in purchasing all three types of kiwi, while only 13.7% of the sample is not interested in purchasing any of the three types of kiwi proposed (Table 8).

These data have been correlated with the purchase of kiwi in general (not necessarily the three types proposed; green kiwi is also included). From the following table, it is evident that the number of proposed kiwi types (yellow, red, and organic) to which respondents answered with 1 (assigning a value of 1 to those who reported that they had purchased one of the indicated kiwifruits or had a strong propensity to purchase them) is positively correlated with the frequency of kiwi purchase (question Q7_1) (frequencies concentrate on the main diagonal of the table) (Table 9). This helps us understand that the inclination to answer 1 to one or more of the proposed kiwi types is correlated with the ATTITUDE TOWARDS KIWI, which we will discuss later, and more generally with the attitude towards fruit consumption (ATTITUDE TOWARDS FRUIT), of which the frequency of kiwi purchase is one revealing aspect of the same attitude level [30] (Appendix A).

#### 3.6.2. Relationship Between Yellow, Red, and Organic Kiwi and Socioeconomic Characteristics

The relationship between the variables YELLOW, RED, and ORGANIC and the socioeconomic characteristics of the interviewees—gender, age groups, level of education, and monthly income class, as well as the country of residence of the interviewees—typically used for market segmentation in marketing activities, has been investigated.

For this purpose, a multivariate logistic model of the following type was considered, where the probability of responding 1 to one of the three variables considered individually is a function of the aforementioned variables. In particular:*P*(*Y* = 1|*X*) = *exp*(*a* + *bX*)/(1 + *EXP*(*a* + *bX*))
where Y is respectively YELLOW, RED, or ORGANIC, and X is a vector of explanatory variables, a: parameter, and b: a vector of parameters. Since the socioeconomic variables are all categorical, we proceeded to construct the following dummy variables:

Furthermore, as is well known, one class of categorical variables should not be included in the model, as it would generate multicollinearity and make it impossible to estimate the model. In this case, the following variables were not included: MALE, AGE < 25, PRIMARY, ITALY, INCOME up to 1000. An individual with these modalities constitutes the reference term (or BASE individual), and the effect of all these modalities is loaded, as known, into the intercept. Therefore, the coefficient of FEMALE will indicate the differential value of being FEMALE compared to the reference individual. Obviously, if some variables are not significant, they should be removed from the model, and the characteristics of the BASE individual would change. If FEMALE is not significant, the base individual does not have gender as a distinctive feature (Table 10).

#### 3.6.3. Results for YELLOW Kiwi

Only individuals with secondary education show a significantly higher propensity to purchase, while in GERMANY, the propensity is lower (always compared to the BASE individual).

Higher propensities for purchase are evident for all income classes except for up to 1500, but the growth is not linear. We observe a peak for the class up to 2500, while for incomes above 3000, the incremental effect appears lower.

##### Methodological Notes: Goodness of Fit of the Models

The previous models have a relatively low goodness of fit and explanatory capacity, as they lack a fundamental variable, namely the ATTITUDE TOWARDS KIWI, which can be measured through the Rasch model, as will be explained in more detail below. The goodness of fit of the above models can be measured using an indicator similar to the R-squared in linear regression, namely the Nagelkerke R2 index, given by the expression:R^2^_N_ = R^2^_CS_/(1 − e^2LL^_0_^/n^)
where:R^2^_CS_ = 1 − e^−2/n (LL^_1_^−LL^_0_^)^
and where LL_1_ refers to the log-likelihood of the model with all explanatory variables, while LL_0_ refers to the log-likelihood of the model without explanatory variables (assuming that the coefficients are all equal to zero). The next table shows the value of this index. As you can see, the values are quite far from the maximum value of 1, so it is worth exploring models with greater predictive capacity. This can be achieved by incorporating a variable into the model that measures ATTITUDE TOWARDS KIWI.

### 3.7. Attitude Towards Fruit and Kiwi Consumption

In this chapter, the results regarding the attitude towards the consumption of fresh fruit in general and kiwi in particular are presented (Appendix A). Firstly, the frequencies of responses to the formulated questions are reported, calculated using descriptive statistical techniques. Percentages were calculated considering the total number of respondents (1202). The items are ordered descending order of positive judgments. It highlights in particular:-The importance of fresh fruit for its nutritional characteristics, so much so that for many, it is essential to eat it every day.-People generally like kiwi (86.6% of respondents), especially for its vitamin C content, but less for its laxative effect.-If available, green kiwi varieties are preferred (72.5%); followed by yellow and red varieties, which are still less known.-Price does not pose an obstacle to fruit purchase; respondents expressed a willingness to pay higher prices than usual to try red kiwi.

#### 3.7.1. Attitude Towards Kiwi

By ATTITUDE TOWARDS KIWI, we mean the level of liking for kiwi in general. This attitude can be measured through the Rasch model applied to those questionnaire items designed as more suitable indicators to represent manifestations of this attitude, along with other indicators that we will see are compatible with the Rasch model itself. It is worth emphasizing that Rasch measures are interval scale, allowing analysis through linear regression models for quantitative data.

The items selected by the Rasch model for measuring ATTITUDE TOWARDS KIWI are listed in Appendix A. The measures of ATTITUDE TOWARDS KIWI of respondents and the so-called “difficulties” of items are compared on the same measurement scale. The higher the value of the difficulty measure, the greater the ATTITUDE TOWARDS KIWI of people who responded to that item with high response values (note that the items have variable scales, but this is not a problem for Rasch models; in fact, the Partial Credit model was used, allowing different scales for each item).

Yet, the order of items is indicative of the scale’s logical consistency: “I like to eat kiwi” (in general K01, the same K06) is at the lowest difficulty levels, and indeed (Appendix A) only 7% of respondents declare an absolute liking for this fruit. It is a question on which it is relatively easy to express oneself with high scale values (1, 2, 3, 4), placing it at the lower difficulty levels.

Just above, we find K14, which is more difficult than K01 because YELLOW kiwi is rarer than green kiwi, which undoubtedly comes to mind for those responding to question K01. Nevertheless, it is a relatively easy item, placing it at around M-S on the difficulty scale (M = 0 and S = 1 approximately). Here, we also find K02 and K03, items highlighting one of the basic reasons for consuming kiwi: its nutritional properties and vitamin C content, fundamental aspects that any advertising campaign aiming to develop ATTITUDE TOWARDS KIWI should focus on.

The knowledge of YELLOW kiwi (K17) also falls at M-S. Considering that 72% declared knowing YELLOW kiwi, if this percentage were higher, this item would be much easier. As the inclination to acquire YELLOW kiwi is a relatively easy item, one way to increase its sales potential is to spread awareness, perhaps emphasizing its higher vitamin C content compared to green ones (Vitamin C 95 mg/100 g versus 95 mg/100 g), which, as seen, is an easy item and the basis on which to encourage consumers to choose this fruit.

Much higher on the scale (M-0.5S), we find K16, which immediately tells us that the organic option does not facilitate kiwi sales as much as the YELLOW alternative. We then find K12 (trying new flavors), not being an easy item, makes us understand that any novelty (e.g., RED kiwi) is unfortunately destined not to be accepted by a portion of consumers precisely due to the difficulty of this item.

Just above the average difficulty level, we find K07, and just above the difficulty level of K19, indicating the lesser spread of knowledge about organic kiwi compared to YELLOW ones. Equally challenging is item K15, indicating that the inclination to purchase RED kiwi is more difficult than YELLOW and ORGANIC ones. Similarly difficult is K08, whose result tells us that it is relatively difficult to choose YELLOW kiwi when all three types are available, especially (considering other responses) the green ones.

At a higher difficulty level (M + 0.5S), items K05 and then K13 are placed. It is interesting to note that K10 (willingness to pay 25% more to taste a RED kiwi) is obviously easier than K11 (willingness to pay 50% more to taste a RED kiwi). Finally, K18 (knowledge of RED kiwi) represents the most difficult item, undoubtedly explaining the greater difficulty in the inclination to purchase this kiwi compared to the other two options (YELLOW and ORGANIC).

Also, items are represented with their individual categories (1, 2, 3, 4, etc.) positioned at the attitude level where the category has its highest probability of occurring among individuals. In particular, the graph highlights the positions of response categories for items K14, K15, and K16, representing declared inclinations for YELLOW, RED, and ORGANIC kiwi, respectively.

As seen in the graph, a high inclination towards RED indicates a very high attitude value and therefore a high difficulty in the occurrence of this event. Next is ORGANIC, followed by YELLOW with lower difficulty compared to the other two. In essence, the graph shows that, in ascending order, YELLOW is easier to like than ORGANIC and then RED.

Those with a very low kiwi attitude (close to −4) express a low inclination for all three types of kiwi (declaring 1 = definitely would not buy).

Overall, the constructed scale appears of good quality, especially considering the good INFIT indices, and all items show a strong positive correlation (PTMEA-CORR) [31] with estimated ATTITUDES (a fundamental requirement of the Rasch model).

In the next paragraph, we will explore how ATTITUDE TOWARDS KIWI can explain the YELLOW, RED, and ORGANIC variables constructed in this paragraph, along with the socioeconomic variables considered earlier.

#### 3.7.2. Attitude Towards YELLOW Kiwi

The results of estimating a multivariate logistic model explaining the probability of YELLOW = 1 (i.e., having purchased yellow kiwi or being definitely or somewhat favorable to their purchase) based on socioeconomic variables and ATTITUDE TOWARDS KIWI estimated with the Rasch model in the preceding paragraph are listed here. For this purpose, it has been transformed into a variable with a mean M = 50 and standard deviation S = 10, essentially providing a scale ranging from 0 = minimum attitude to 100 = maximum attitude (results obtained through non-standardized measures are identical except for the coefficient value of the ATTITUDE TOWARDS KIWI variable).

ATTITUDE TOWARDS KIWI is decidedly significant, and all other variables lose their significance except for the SPAIN variable. Therefore, we have removed all non-significant variables from the model and estimated a new model with only significant variables: this model has an R2 = 0.575, well above what could be achieved with only socioeconomic variables. The coefficient of the ATTITUDE TOWARDS KIWI variable is +0.315 (*p*-value = 0.000), while SPAIN has a coefficient of +0.606 (*p*-value = 0.111, at the limits of usual significance).

The theoretical probabilities of YELLOW = 1 for different values of the ATTITUDE TOWARDS KIWI variable (BASE individual, which is everything but Spanish) and the difference in being Spanish compared to the base individual are also reported here. At the average level of ATTITUDE M = 50, the probability of YELLOW = 1 is 89%, and being Spanish at this level of attitude increases this probability by 3%. The probability at M-2S is very low (1%), while at M-S, it is 25% (with an 8% increase for a BASE individual with Spanish nationality).

In general, therefore, the inclination to purchase YELLOW kiwi is determined by ATTITUDE TOWARDS KIWI and nothing else, except for being Spanish, which increases this inclination by a few percentage points.

#### 3.7.3. Attitude Towards RED Kiwi

The outcomes of estimating a multivariate logistic model explaining the probability of RED = 1 (i.e., having purchased red kiwi or being definitely or somewhat favorable to their purchase) based on socioeconomic variables and ATTITUDE TOWARDS KIWI estimated with the Rasch model, transformed into a variable with a mean M=50 and standard deviation S=10, are reported as follows (Table 11).

ATTITUDE TOWARDS KIWI is decidedly significant, and all other variables lose their significance, except for the variables FEMALE, SPAIN, and FRANCE. Therefore, we have removed all non-significant variables from the model and estimated a new model with only significant variables: this model has an R^2^ = 0.465, well above what could be achieved with only socioeconomic variables. The coefficient of the ATTITUDE TOWARDS KIWI variable is +0.243 (*p*-value = 0.000), while FEMALE is −0.365 (*p*-value = 0.018), SPAIN has a coefficient of −0.672 (*p*-value = 0.000), and FRANCE is −0.665 (*p*-value = 0.000).

The values of the theoretical probabilities of RED = 1 for different values of the ATTITUDE TOWARDS KIWI variable (BASE individual, which is male, not Spanish, nor French) and the difference in being characterized by one of the significant explanatory variables (in addition to attitude) compared to the base individual are also reported. At the average level of ATTITUDE M = 50, the probability of RED = 1 is 52% (much lower than 89% for YELLOW), and being FEMALE at this level of attitude reduces this probability by −9%. Being Spanish or French reduces it by −16%, and being both female and Spanish reduces it by −24%. To have a very high probability of RED = 1, one must be at levels of ATTITUDE towards KIWI of M + S (92%).

In general, therefore, the inclination to purchase RED kiwi is also determined by ATTITUDE TOWARDS KIWI, and to a greater extent than for YELLOW kiwi. Only at high levels of ATTITUDE are high probabilities of RED = 1 observed. Additionally, in the case of RED kiwi, gender and Spanish or French nationality also play a significant role: all three characteristics reduce the inclination to purchase RED kiwi.

Lastly, our analysis had previously revealed that the linear regression model which assessed the impact of gender, age, education level, and income on kiwi preference, yielded a very low R^2^ value. This indicates that socio-demographic characteristics alone do not adequately explain consumer attitudes toward kiwi. In contrast, the upcoming results from Table 11 show that the propensity to purchase different types of kiwi is strongly influenced by the constructed variable measuring attitude toward kiwi, while the influence of socio-demographic factors remains limited. This highlights the central role of psychographic variables in shaping consumer behavior. Consequently, future marketing research should prioritize the development of explanatory constructs of this nature—ensuring they are composed of items that meet the requirements of the Rasch model, which guarantees specific objectivity, comparability over time and across contexts, and invariance across populations.

#### 3.7.4. Attitude Towards Organic Kiwi

The result of estimating a multivariate logistic model, explaining the probability of ORGANIC = 1 (i.e., having purchased red kiwi or being definitely or somewhat favourable to their purchase) based on socioeconomic variables and ATTITUDE TOWARDS KIWI estimated with the Rasch model, transformed into a variable with a mean M = 50 and standard deviation S = 10, is listed. ATTITUDE TOWARDS KIWI is significantly significant, but unlike the other two variables YELLOW and ORGANIC, several socioeconomic explanatory variables maintain their significance (AGE55+, TERTIARY, SPAIN, FRANCE), with income this time showing a significant effect.

We have therefore removed all non-significant variables from the model (except some related to income) and estimated a new model with only significant variables: this model has an R^2^ = 0.553, significantly higher than what could be achieved with only socioeconomic variables. The coefficient of the ATTITUDE TOWARDS KIWI variable is +0.275 (*p*-value = 0.000), while AGE55+ has a coefficient of −0.588 (*p*-value = 0.001), TERTIARY has a positive coefficient of +0.482 (*p*-value = 0.013), SPAIN has a coefficient of −1.487 (*p*-value = 0.000), and FRANCE is −1.273 (*p*-value = 000). The coefficients of income from 200 euros and above are almost all significant, ranging from +0.432 to +0.538.

Furthermore, the values of the theoretical probabilities of ORGANIC = 1 for different values of the ATTITUDE TOWARDS KIWI variable (BASE individual, not French or Spanish, under 55 years old, with education below the third level and income up to 1500 euros) and the difference in being characterized by one of the significant explanatory variables (in addition to attitude) compared to the base individual are explained. At the average level of ATTITUDE M = 50, the probability of ORGANIC = 1 is 80%. Being 55+ reduces it by −11%, having TERTIARY education increases it by +7%, being Spanish or French reduces it by −33% and −27%, respectively, and having an income of up to 2000 euros and above increases it by +7%.

Therefore, even in the case of organic kiwi, the attitude towards purchase is determined by ATTITUDE TOWARDS KIWI. The propensity decreases in the case of age over 55, and if the individual is Spanish or French; instead, it increases in the case of tertiary education and higher income levels. In particular, in the case of a BASE individual but AGE55+ and Spanish, the overall reduction is −47%, while BASE but with tertiary education and income up to 2000 euros results in an increase of 12%.

#### 3.7.5. Explanatory Model of Attitude Towards Kiwi

Since ATTITUDE TOWARDS KIWI is a fundamental factor in the propensity to purchase all three types of proposed kiwi, it is appropriate to investigate the explanatory factors of this variable, which we have identified in the usual socioeconomic factors and ATTITUDE TOWARDS FRUIT (in general), the latter variable constructed from the Rasch model applied to questionnaire data.

In particular, to measure this variable, the Rasch model has selected some of the items (Table 12 and Table 13). In this case, too, the order of the items is indicative of the goodness of the scale identified from the point of view of its logical consistency: F04 Fresh fruit is a very healthy product, and F01 I like to eat fresh fruit are at the lowest levels of the difficulty scale of the items selected by the model. Just above, we find F05 and F03, which highlight the reasons for consuming fresh fruit (nutrients that are good for health); then, there is a difficulty jump with item F02 (−0.39): only 56% of those surveyed totally agree that it is essential to eat fresh fruit every day. Above this item and essentially at the average difficulty level (zero by definition), we find F12, F13, and F15, which indicate respondents’ preferences for kiwi. This means that only half of the population likes kiwi; in fact, to the question F12 I like to eat kiwi, only 57% answered “4 = I totally agree.” Moving up the difficulty scale (Table 12), we find F14, which highlights one of the reasons for eating kiwi (Vitamin C), one of the strong points that should be the focus of any advertising campaign related to this fruit, as already highlighted in the case of attitude towards kiwi. Essentially at the same difficulty level, we find the preference for kiwi as an ingredient in other food preparations (F07). At a higher level of difficulty, we find the frequency of banana consumption (F11), which has a difficulty of 0.50: it is noted that the same question for kiwi (F10) has a much higher difficulty of 1.66. The same applies to the questions about the frequency of purchasing bananas (F09) and kiwi (F08). Finally, it is observed that F08, the frequency of kiwi purchase, represents the item with the highest difficulty [32].

The variable ATTITUDE TOWARDS FRUIT was then calculated for each individual through the Rasch model and added to the socioeconomic variables in a regression model explaining ATTITUDE TOWARDS KIWI. Below is the regression model with only significant explanatory variables (Table 14). As seen, the model has an R^2^ = 0.529 (the R^2^ value with only socioeconomic variables would be much smaller, around 0.05).

The significant explanatory variables for ATTITUDE TOWARDS KIWI are ATTITUDE TOWARDS FRUIT, which has a positive but nonlinear effect, highlighted in the graph (Figure 7). Holding ATTITUDE TOWARDS FRUIT constant, a reduction in ATTITUDE TOWARDS KIWI is observed for those over 55 (−0.235) and individuals of German nationality (−0.122). We observe that the estimated ATTITUDE is not categorical, but an INTERVAL measure by the property of the Rasch model.

#### 3.7.6. Kiwi Attitude: Summary of Results

As observed, YELLOW, RED, and ORGANIC are directly influenced by ATTITUDE TOWARDS KIWI and some socioeconomic variables (direct effects), along with other unknown factors. ATTITUDE TOWARDS KIWI, in turn, is explained by ATTITUDE TOWARDS FRUIT and some socioeconomic variables (indirect effects) (Appendix A). The following graphical diagrams summarize the obtained results.

In all cases, we found a positive effect of ATTITUDE TOWARDS KIWI on the three variables YELLOW, RED, and ORGANIC, and a positive effect of ATTITUDE TOWARDS FRUIT on ATTITUDE TOWARDS KIWI. Therefore, we will focus on the direct and indirect effects of socioeconomic variables.

YELLOW: being SPAIN has a direct positive effect, while being AGE55+ and GERMANY has an indirect negative effect. To increase the consumption of YELLOW kiwi, it is necessary to increase ATTITUDE TOWARDS KIWI in the elderly and in Germany; moreover, campaigns in Italy, Germany, and France specifically regarding YELLOW kiwi are needed (Figure 8).

RED: being FEMALE, SPAIN, or FRANCE has a direct negative effect, while being AGE55+ and GERMANY has an indirect negative effect. To increase the consumption of RED kiwi, it is necessary to increase ATTITUDE TOWARDS KIWI in the elderly and in Germany; moreover, campaigns in Spain and France specifically regarding RED kiwi, along with interventions targeting the female population, are required (Figure 9).

ORGANIC: being AGE 55+, SPAIN, or FRANCE has a direct negative effect, while TERTIARY and > 1500 EURO have a direct positive effect; being AGE55+ and GERMANY has an indirect negative effect. To increase the consumption of ORGANIC kiwi, it is necessary to increase ATTITUDE TOWARDS KIWI in the elderly and in Germany; moreover, targeted campaigns for ORGANIC in Spain and France, targeting the elderly population with educational levels below tertiary and low incomes, are needed (Figure 10).

## 4. Concluding Remarks

This study investigated consumer attitudes toward fruit in general, kiwi in general, and three specific types of kiwi (yellow, red, and organic), using a comprehensive set of attitudinal items (e.g., motivations for consumption, willingness to pay, interest in new foods, etc.). The results clearly show that the likelihood of purchasing yellow, red, and organic kiwis is significantly influenced by the general attitude towards kiwi, which itself is strongly shaped by the broader attitude towards fruit. This cascade effect highlights the foundational role of fruit perception in shaping consumer behaviour toward more specific products.

Moreover, the multiple linear regression analysis identified several socio-demographic variables with a statistically significant effect. Age groups 25–34, 35–44, 45–54, and 55+ were all positively associated with more favourable attitudes towards fruit, while being male, under 25, having a low income, low educational attainment, and being of German nationality negatively affected such attitudes. These findings suggest key segments where targeted marketing and educational campaigns could have the greatest impact.

Based on these insights, the following data-driven marketing recommendations emerge:To promote yellow, red, and organic kiwis, efforts should focus on enhancing kiwi attitudes, particularly among elderly consumers and German respondents, by emphasizing benefits from the more challenging attitudinal items (e.g., sleep quality improvement, digestive health, versatility in recipes).For yellow kiwi, intensified communication in Italy, Germany, and France is needed, given lower purchase intentions there compared to Spain, where readiness to buy is higher.For red kiwi, promotional actions should be prioritized in Spain and France, especially targeting women.Regarding organic kiwi, campaigns in Spain and France should focus on older individuals with low income and lower educational levels.

An additional implication of our findings is the importance of fostering a positive general attitude toward fruit, particularly in groups where this is currently low. By increasing fruit appreciation in young males with low socioeconomic status, marketers could indirectly raise kiwi consumption, especially of the more niche types.

While this study provides robust evidence from over 1000 respondents, its conclusions are naturally bound by the sample’s demographic and geographic scope. To deepen the understanding of global kiwi consumption patterns, future studies should include a wider range of cultural and socioeconomic contexts, as well as emerging factors such as environmental values, sustainability concerns, and changing health trends.

In summary, this work contributes to the literature by empirically demonstrating the layered structure of attitudes from fruit to specific kiwi types, and by offering clear, evidence-based guidance for targeted marketing strategies. Strengthening attitudes towards fruit represents a promising lever for influencing consumer decisions in the fruit sector, and the study sets the stage for further exploration of how behavioural drivers can be harnessed to promote healthier and more sustainable dietary choices.

## Figures and Tables

**Figure 1 foods-14-02683-f001:**
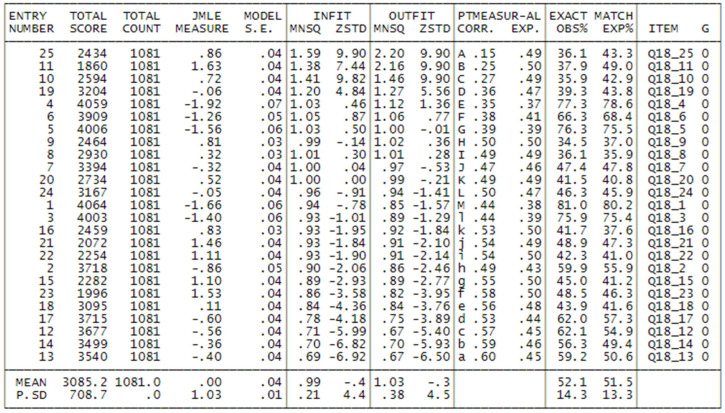
Items of the questionnaire in misfit order. Item measures are expressed in Rasch Unit. In the context of Rasch models, particularly the Rating Scale Model (RSM), the INFIT and OUTFIT statistics are fit indices used to evaluate how well the observed data conform to the expectations of the Rasch model. Acceptable Ranges for INFIT and OUTFIT (Rating Scale Model). 👉 0.7 to 1.3 for most applications. 👉 0.8 to 1.2 for high-stakes testing or strict measurement.

**Figure 2 foods-14-02683-f002:**
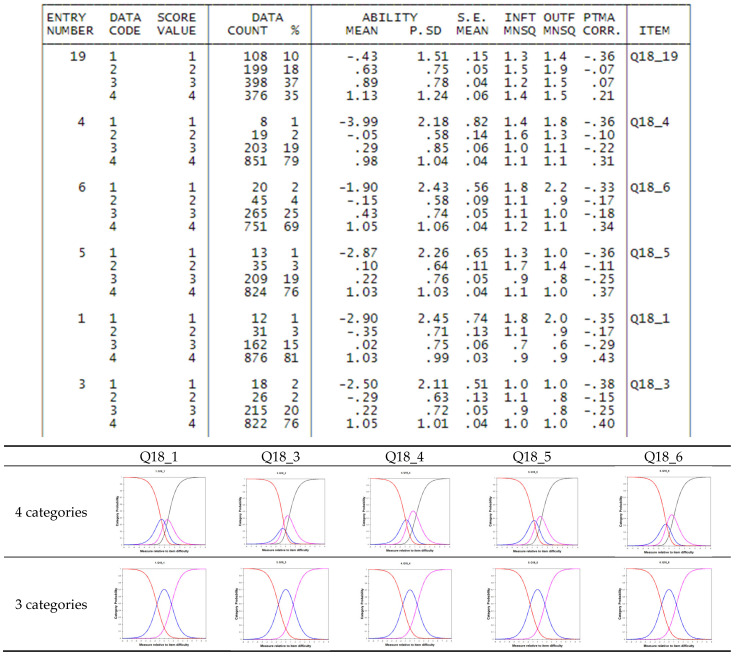
Point measure correlations between scores and measure and Category characteristic curves, before and after collapsing.

**Figure 3 foods-14-02683-f003:**
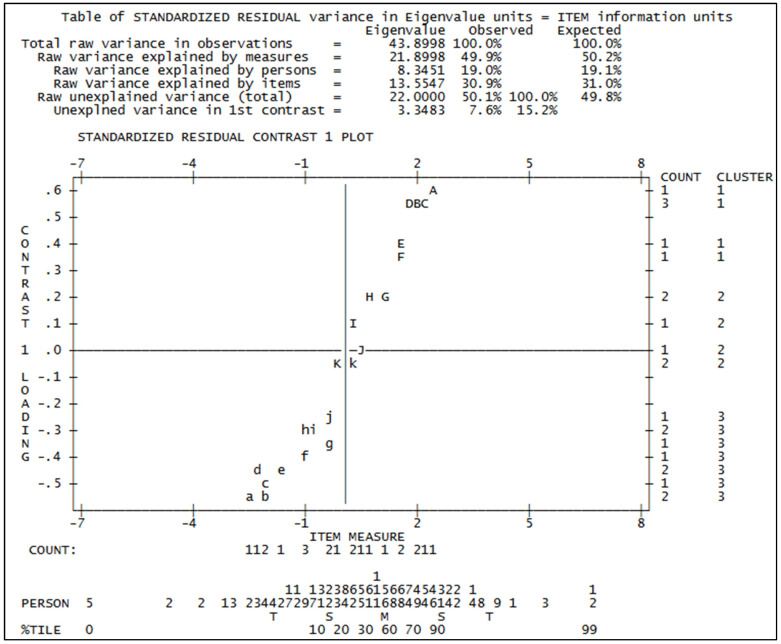
Analysis of uni-dimensionality.

**Figure 4 foods-14-02683-f004:**
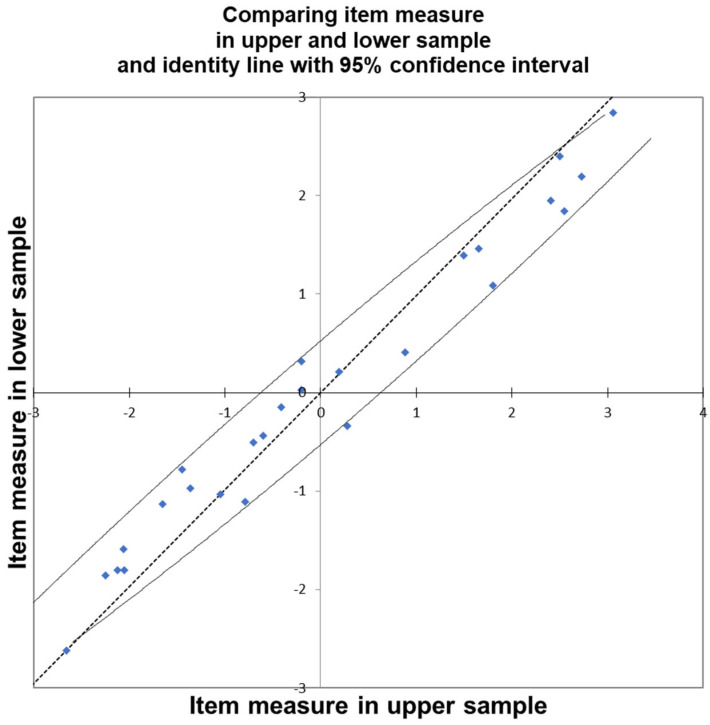
Item estimated with the upper and lower sample, identity line and 95% confidence interval.

**Figure 5 foods-14-02683-f005:**
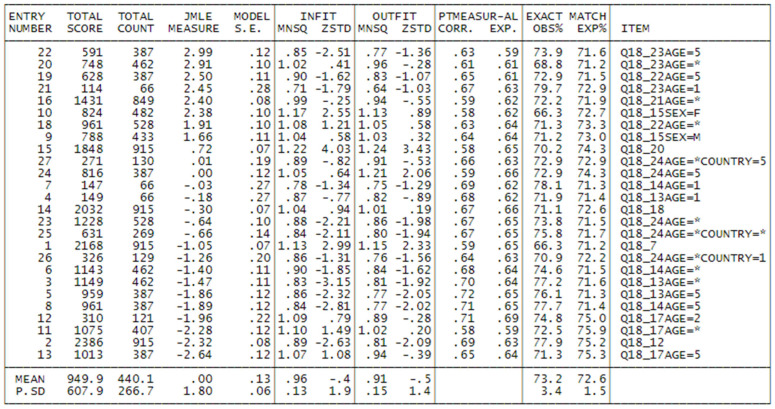
Item measure and fit for kiwi attitude from the final model.

**Figure 6 foods-14-02683-f006:**
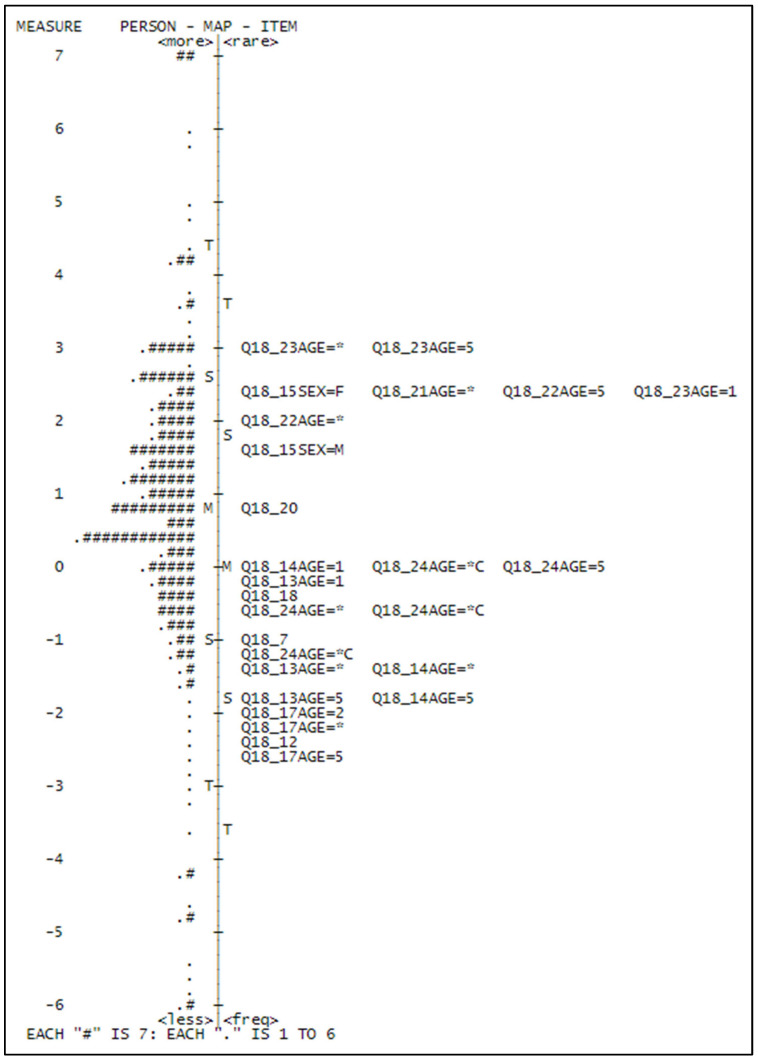
Wright map: average difficulties of the items (**right**) contrasted to the distribution of person measures (**left**).

**Figure 7 foods-14-02683-f007:**
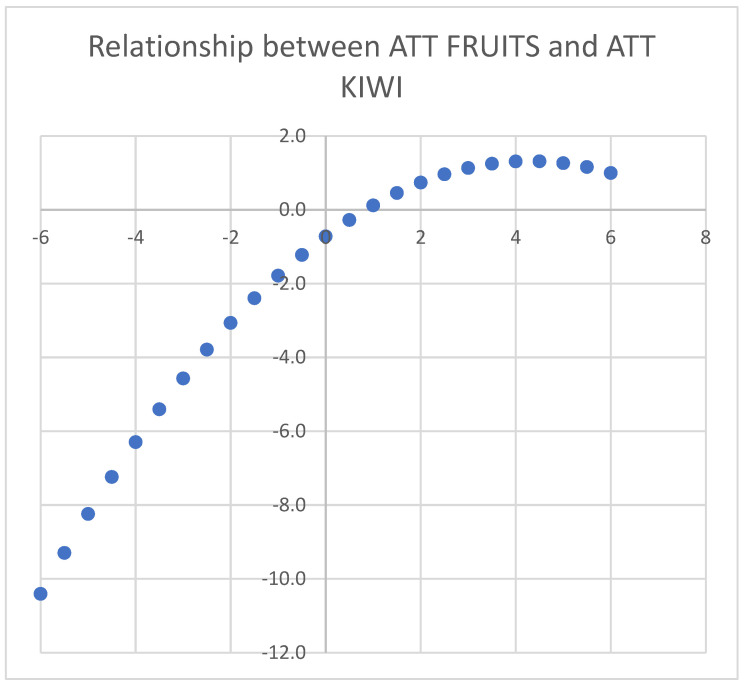
The relationship Y = f(X) between attitudes towards kiwi (Y) and attitudes towards fruit (X).

**Figure 8 foods-14-02683-f008:**
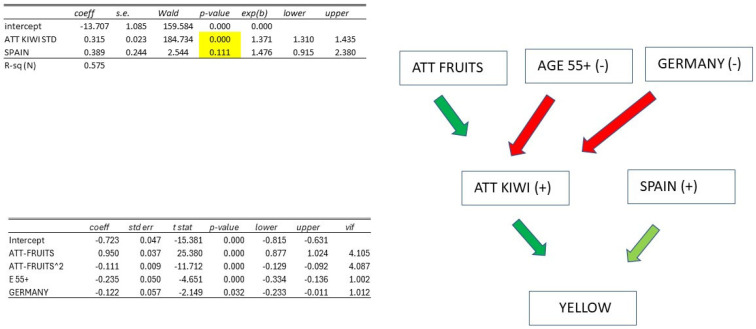
Summary of attitudes towards yellow kiwi.

**Figure 9 foods-14-02683-f009:**
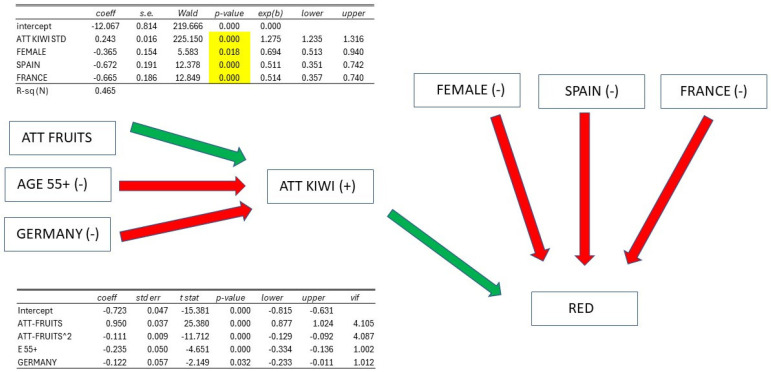
Summary of attitudes towards red kiwi.

**Figure 10 foods-14-02683-f010:**
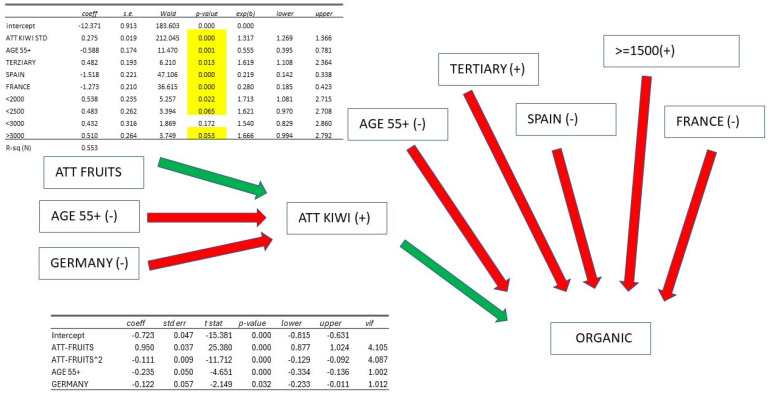
Summary of attitudes towards organic kiwi.

**Table 1 foods-14-02683-t001:** Relative frequency distributions of the key variables considered in this study.

**COUNTRY**	**COMPLETE SAMPLE**	**AFTER EXCLUSIONS**	**DIFFERENCES**
1	25.0%	23.0%	1.9%
3	25.0%	25.2%	−0.1%
4	25.0%	26.1%	−1.1%
5	25.0%	25.7%	−0.7%
**TOTAL**	100.0%	100.0%	0.0%
**SEX**	**COMPLETE SAMPLE**	**AFTER EXCLUSIONS**	**DIFFERENCES**
1	47.8%	48.1%	−0.3%
2	52.0%	51.7%	0.3%
3	0.2%	0.2%	0.0%
**TOTAL**	100.0%	100.0%	0.0%
**AGE**	**COMPLETE SAMPLE**	**AFTER EXCLUSIONS**	**DIFFERENCES**
1	9.2%	9.3%	−0.1%
2	15.4%	15.4%	0.0%
3	18.7%	18.7%	0.0%
4	18.1%	18.1%	0.1%
5	38.5%	38.5%	0.0%
**TOTAL**	100.0%	100.0%	0.0%
**DEEGREE**	**COMPLETE SAMPLE**	**AFTER EXCLUSIONS**	**DIFFERENCES**
1	1.8%	1.9%	0.0%
2	15.8%	15.9%	0.0%
3	19.5%	18.8%	0.7%
4	29.0%	29.1%	−0.2%
5	15.5%	15.6%	−0.1%
6	11.0%	10.9%	0.1%
7	3.7%	3.9%	−0.2%
8	3.8%	4.0%	−0.2%
**TOTAL**	100.0%	100.0%	0.0%
**INCOME**	**COMPLETE SAMPLE**	**AFTER EXCLUSIONS**	**DIFFERENCES**
1	18.1%	18.2%	−0.2%
2	20.0%	20.1%	−0.1%
3	20.4%	20.2%	0.2%
4	14.6%	14.8%	−0.1%
5	9.4%	8.9%	0.5%
6	17.6%	17.7%	−0.2%
**TOTAL**	100.0%	100.0%	0.0%

**Table 2 foods-14-02683-t002:** Items and descriptions.

LOADING		Item	Description
**POSITIVE**	**A**	**Q18_23**	I am willing to pay 50% more than the usual price to taste red kiwifruit.
	**C**	**Q18_21**	If green, yellow, and red kiwifruit are available, I choose the RED ones.
	**B**	**Q18_15**	I eat kiwifruit (after dinner) because it improves my sleep quality.
	**D**	**Q18_22**	I am willing to pay 25% more than the usual price to taste a red kiwifruit.
**NEGATIVE**	**c**	**Q18_5**	Fresh fruit provides a variety of nutrients.
	**a**	**Q18_1**	I like eating fresh fruit.
	**d**	**Q18_4**	Fresh fruit is a very healthy product.
	**b**	**Q18_3**	I eat fresh fruit because it is good for my health.

**Table 3 foods-14-02683-t003:** Items selected for measuring kiwi attitude.

Q18_7 I like eating fresh fruit as an ingredient in cakes, fruit salads, smoothies, etc.
Q18_12 I like eating kiwifruit.
Q18_13 I eat kiwifruit for its remarkable nutritional properties.
Q18_14 I eat kiwifruit for its high vitamin C content.
Q18_15 I eat kiwifruit (after dinner) because it improves my sleep quality.
Q18_16 I eat kiwifruit for its laxative effect.
Q18_17 I like eating kiwifruit as it is.
Q18_18 I like eating kiwifruit as an ingredient in cakes, fruit salads, smoothies, etc.
Q18_19 If green, yellow, and red kiwifruit are available, I choose the GREEN ones.
Q18_20 If green, yellow, and red kiwifruit are available, I choose the YELLOW ones.
Q18_21 If green, yellow, and red kiwifruit are available, I choose the RED ones.
Q18_22 I am willing to pay 25% more than the usual price to taste a red kiwifruit.
Q18_23 I am willing to pay 50% more than the usual price to taste a red kiwifruit.
Q18_24 I like experimenting with new foods and flavours, even if I don’t know them.

**Table 4 foods-14-02683-t004:** Synthesis of indexes for different fits of the Rasch model to the data.

Indexes	Preliminary Model	Final Model with Diff
**Personal reliability**	0.86	0.87
**Separation**	2.52	2.64
**Item reliability**	1	0.99
**Separation**	18.62	11.95
**Category infit** **1**	0.98	1.01
**2**	0.91	1.00
**3**	1.02	0.99
**Andrich Threshold**	−2.04	−2.29
	2.04	2.29
**Eigenvalue 1st contrast**	2.13	2.30
**Deattenuated correlation 1–3**	0.74	0.89
**Items at extremes**	Q18_23	Q18_24AGE
	Q18_13	Q18_14AGE
**Misfitting item**	Q18_19 (excluded)	Q18_21AGE=1 (excluded)
**Max item Infit/Outfit**	1.32/1.29	1.22/1.24
**Persons selected**	933	915

**Table 5 foods-14-02683-t005:** Differential item functioning: items with significant t-values for different person characteristics.

Item	Sex	Age	Degree	Country
Q18_7	NO	NO	NO	NO
Q18_12	NO	NO	NO	NO
Q18_13	NO	YES (1.5)	YES (1)	NO
Q18_14	NO	YES (1.5)	NO	NO
Q18_15	YES	NO	YES (8)	YES (5)
Q18_17	NO	YES (2.5)	NO	NO
Q18_18	NO	NO	NO	NO
Q18_20	NO	NO	NO	NO
Q18_21	NO	YES (1)	NO	NO
Q18_22	NO	YES (5)	YES (1)	NO
Q18_23	NO	YES (1.5)	NO	NO
Q18_24	NO	YES (5)	NO	YES (1.5)

**Table 6 foods-14-02683-t006:** Multiple linear regression model explaining attitudes towards fruits.

Overall Fit								
Multiple R	0.231		AIC	979.53				
R Square	0.054		AICc	980.10				
Adjusted R Square	0.040		SBC	1059.49				
Standard Error	1.553							
Observations	1094							
ANOVA				Alpha	0.05			
	*df*	*SS*	*MS*	*F*	*p-value*	*sig*		
Regression	15	147.1642	9.810945	4.065969	2.86 × 10^−7^	yes		
Residual	1078	2601.151	2.412941					
Total	1093	2748.315						
	*coeff*	*std err*	*t stat*	*p-value*	*lower*	*upper*	*vif*	
Intercept	0.746	0.238	3.132	0.002	0.278	1.213		
FEMALE	0.169	0.096	1.761	0.078	−0.019	0.358	1.045	
AGE 25–34	0.483	0.206	2.343	0.019	0.079	0.888	2.435	
AGE 35–44	0.465	0.198	2.351	0.019	0.077	0.854	2.766	
AGE 45–54	0.422	0.201	2.096	0.036	0.027	0.818	2.717	
AGE 55+	0.554	0.185	3.002	0.003	0.192	0.916	3.697	
SECONDARY	0.223	0.135	1.660	0.097	−0.041	0.487	2.051	
TERZIARY	0.074	0.151	0.489	0.625	−0.222	0.369	2.286	
SPAIN	0.094	0.137	0.685	0.494	−0.176	0.364	1.613	
FRANCE	0.049	0.138	0.354	0.724	−0.222	0.319	1.599	
GERMANY	−0.414	0.137	−3.018	0.003	−0.684	−0.145	1.625	
<1500	−0.160	0.154	−1.034	0.301	−0.463	0.143	1.749	
<2000	0.212	0.156	1.357	0.175	−0.095	0.518	1.763	
<2500	0.467	0.169	2.758	0.006	0.135	0.800	1.635	
<3000	0.523	0.191	2.730	0.006	0.147	0.898	1.454	
>3000	0.465	0.167	2.790	0.005	0.138	0.792	1.802	

**Table 7 foods-14-02683-t007:** Which of the following KIWI do you know?

	Italy	Spain	France	Germany	Total
Yellow kiwi	61.3	82.4	77.4	68	72.3
Red kiwi	17.7	13.6	11	6.7	12.2
Organic kiwi	61	42.9	48.2	64	54
None of the previous	12.7	8.6	10.3	12.7	11.1
**Total**	**100**	**100**	**100**	**100**	**100**

Note: Respondents: all interviewees (total 1202). Multiple responses were possible.

**Table 8 foods-14-02683-t008:** Frequency distribution (absolute and percentage) of the total number of 1s expressed for YELLOW, RED, and ORGANIC.

Kiwi Number = 1	Number	%
3	452	37.6
2	315	26.2
1	270	22.5
0	165	13.7
**Total**	**1202**	**100**

**Table 9 foods-14-02683-t009:** Double frequency distribution of kiwi purchase and the number of KIWI = 1.

	Number of KIWI = 1				
Frequency of purchasing kiwi	0	1	2	3	Total
Every day	1	7	18	27	53
4/5 times per week	1	5	15	43	64
2/3 times per week	8	14	50	103	175
1 time per week	20	65	93	123	301
2/3 times per month	13	51	51	86	201
1 time per month	12	62	42	46	162
Less than 1 time per month	46	54	40	24	161
Never	67	12	3	0	85
**Full total**	**165**	**270**	**315**	**452**	**1202**

**Table 10 foods-14-02683-t010:** Dummy variables and their values.

Dummy	Values
Female	1 if female, 0 otherwise
Age 25–34	1 if the age group is 25–34, 0 otherwise. The same for the other age groups
Age 35–44	
Age 45–54	
Age 55+	
Secondary	1 if the educational qualification is secondary (professional or secondary), 0 otherwise
Tertiary	1 if the educational qualification is higher than secondary, 0 otherwise
<1500	
<2000	
<2500	1 if the income class is the one indicated, 0 otherwise
<3000	
>3000	
Spain	1 if the nationality is Spain, 0 otherwise
France	2 if the nationality is France, 0 otherwise
Germany	3 if the nationality is Germany, 0 otherwise

**Table 11 foods-14-02683-t011:** The multivariate logistic regression model for the probability of RED = 1 including all variables.

	*coeff*	*s.e.*	*Wald*	*p-Value*	*exp(b)*	*Lower*	*Upper*
Intercept	−11.848	0.903	171.998	0.000	0.000		
ATT KIWI STD	0.244	0.016	218.704	0.000	1.276	1.235	1.318
FEMALE	−0.412	0.159	6.732	0.009	0.663	0.485	0.904
AGE 25–34	0.121	0.324	0.140	0.708	1.129	0.598	2.130
AGE 35–44	−0.129	0.314	0.169	0.681	0.879	0.475	1.627
AGE 45–54	−0.110	0.315	0.123	0.726	0.896	0.483	1.659
AGE 55+	−0.283	0.288	0.965	0.326	0.753	0.428	1.326
SECONDARY	−0.241	0.222	1.184	0.277	0.786	0.509	1.213
TERZIARY	−0.262	0.249	1.114	0.291	0.769	0.472	1.252
SPAIN	−0.658	0.227	8.397	0.004	0.518	0.332	0.808
FRANCE	−0.613	0.222	7.631	0.006	0.542	0.351	0.837
GERMANY	0.055	0.223	0.061	0.805	1.057	0.682	1.636
<1500	0.120	0.264	0.208	0.648	1.128	0.673	1.892
<2000	0.192	0.263	0.534	0.465	1.212	0.724	2.029
<2500	0.009	0.281	0.001	0.975	1.009	0.582	1.748
<3000	−0.177	0.315	0.314	0.575	0.838	0.452	1.554
>3000	0.148	0.282	0.275	0.600	1.159	0.667	2.014

**Table 12 foods-14-02683-t012:** Items for measuring attitudes towards fruits.

Name	Entry	Code	Description	Value			
F01	31	Q18_1	I enjoy eating fresh fruit.	1 = completely disagree	2	3	4 = completely agree
F02	32	Q18_2	Eating fresh fruit every day is essential for me.	1 = completely disagree	2	3	4 = completely agree
F03	33	Q18_3	I eat fresh fruit because it’s good for my health	1 = completely disagree	2	3	4 = completely agree
F04	34	Q18_4	Fresh fruit is a very healthy product.	1 = completely disagree	2	3	4 = completely agree
F05	35	Q18_5	Fresh fruit provides a multitude of nutrients.	1 = completely disagree	2	3	4 = completely agree
F06	36	Q18_6	I enjoy eating fresh fruit as it is, without any special transformations.	1 = completely disagree	2	3	4 = completely agree
F07	37	Q18_7	I enjoy eating fresh fruit as an ingredient in cakes, fruit salads, smoothies, etc.	1 = completely disagree	2	3	4 = completely agree
F08	39	Q7_1BIS	How often do you buy kiwi?	1 = less often or never 2 = 1 to 3 times a month 3 = 1 to 3 times a week 4 = at least 4–5 times a week			
F09	40	Q7_2BIS	How often do you buy bananas?	1 = less often or never 2 = 1 to 3 times a month 3 = 1 to 3 times a week 4 = at least 4–5 times a week			
F10	41	Q10_1BIS	Do you eat kiwi? If yes, how often do you eat them?	1 = less often or never 2 = 1 to 3 times a month 3 = 1 to 3 times a week 4 = at least 4–5 times a week			
F11	42	Q10_2BIS	Do you eat bananas? If yes, how often do you eat them?	1 = less often or never 2 = 1 to 3 times a month 3 = 1 to 3 times a week 4 = at least 4–5 times a week			
F12	43	Q18_12	I like eating kiwi.	1 = completely disagree	2	3	4 = completely agree
F13	45	Q18_13	I eat kiwi for its remarkable nutritional properties.	1 = completely disagree	2	3	4 = completely agree
F14	46	Q18_14	I eat kiwi for its high vitamin C content.	1 = completely disagree	2	3	4 = completely agree
F15	47	Q18_17	I like eating kiwi as it is.	1 = completely disagree	2	3	4 = completely agree
F16	48	Q18_18	I enjoy eating kiwi as an ingredient in cakes, fruit salads, smoothies, etc.	1 = completely disagree	2	3	4 = completely agree

**Table 13 foods-14-02683-t013:** Difficulty of items measuring attitudes towards fruits.

Name	Description	Difficulty	S.E.	Infit	Ptmea-Corr
F08	Do you buy kiwi regularly? Do you eat kiwi? If yes, how often?	2.24	0.05	1.12	0.64
F10	How often do you buy bananas? Do you eat bananas? If yes, how often?	1.66	0.04	1.03	0.67
F09	I like using kiwi as an ingredient in cakes, fruit salads, smoothies, etc.	0.89	0.05	1.24	0.52
F16	I like eating fresh fruit as an ingredient in cakes, fruit salads, smoothies, etc.	0.82	0.04	1.08	0.63
F11	I eat kiwi for its high vitamin C content.	0.50	0.05	1.23	0.55
F07	I eat kiwi for its remarkable nutritional properties.	0.24	0.05	1.23	0.55
F14	I like eating kiwi.	0.20	0.05	0.81	0.66
F13	I like eating kiwi as it is.	0.14	0.05	0.75	0.67
F12	For me, it’s essential to eat fresh fruit every day.	−0.06	0.05	0.74	0.65
F15	I like eating fresh fruit as it is, without any special transformations.	−0.13	0.05	0.84	0.62
F02	I eat fresh fruit because it’s good for my health.	−0.39	0.05	0.92	0.61
F06	Fresh fruit provides a multitude of nutrients.	−0.97	0.06	1.07	0.50
F03	I like eating fresh fruit.	−1.11	0.06	0.91	0.54
F05	Fresh fruit is a very healthy product.	−1.20	0.06	1.06	0.50
F01	Do you buy kiwi regularly? Do you eat kiwi? If yes, how often?	−1.36	0.06	0.88	0.53
F04	How often do you buy bananas? Do you eat bananas? If yes, how often?	−1.47	0.07	1.12	0.47

**Table 14 foods-14-02683-t014:** Multiple linear regression model explaining attitudes towards kiwi consumption.

Overall Fit							
Multiple R	0.727		AIC	−480.476			
R Square	0.529		AICc	−480.392			
Adjusted R Square	0.527		SBC	−455.927			
Standard Error	0.785						
Observations	1002						
ANOVA				Alpha	0.05		
	*df*	*SS*	*MS*	*F*	*p-value*	*sig*	
Regression	4	689.799	172.450	279.947	2.6 × 10^−161^	yes	
Residual	997	614.160	0.616				
Total	1001	1303.959					
	*coeff*	*std err*	*t stat*	*p-value*	*lower*	*upper*	*vif*
Intercept	−0.723	0.047	−15.381	0.000	−0.815	−0.631	
ATT-FRUITS	0.950	0.037	25.380	0.000	0.877	1.024	4.105
ATT-FRUITS^2	−0.111	0.009	−11.712	0.000	−0.129	−0.092	4.087
E 55+	−0.235	0.050	−4.651	0.000	−0.334	−0.136	1.002
GERMANY	−0.122	0.057	−2.149	0.032	−0.233	−0.011	1.012

## Data Availability

The original contributions presented in the study are included in the article/Appendix A. Further inquiries can be directed to the corresponding author.

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
