# Peer review of "Using the Rasch Model to Understand Consumers’ Behaviour in Buying Kiwifruits"

_foods, 2025, doi:10.3390/foods14152683_

Round 1
Reviewer 1 Report
Comments and Suggestions for Authors
Please find attached reviewers comment.

Author Response
Please check the attached file. Many thanks

Reviewer 2 Report
Comments and Suggestions for Authors
I have carefully reviewed this manuscript, and the article applies the Rasch model to analyze consumers' attitudes towards kiwifruit with different flesh characteristics. I agree with the significance of the manuscript proposed by the author, but there are still many issues that need to be addressed. The discussions provided by the author are rough and incomplete. This manuscript needs a major revision before publication and my general comments/suggestions are as follows.
- The authors state that studies not using the Rasch model may produce‘not fully acceptable’ results due to a lack of specific objectivity (Lines 118–120). While the Rasch model indeed provides valuable measurement properties, this statement may be overly absolute. At the same time, the advantages of this study should be highlighted in comparison to the literature.
- The description in the Methods section needs to be more concise and objective, for example, the wording in line148 should be further adjusted. In addition, the concluding paragraph lacks a systematic summary of method selection (lines 280-284).
- Some tables in the text are difficult to read, it is recommended to improve the quality of the tables.Please optimize according to the requirements of the journal.
- The justification for removing the 3 items at Line 319 requires strengthening.
- In the multiple linear regression model in Table 9, the adjusted R squared is only 0.04, which is not common. The article fails to assess the validity of the model's explanatory power or delve into the reasons behind this value.
- The marketing recommendations proposed in the conclusion section can provide further data support. Conclusions need more in it, as it’s more of an afterthought. The authors are suggested to highlight important findings and include afterthought of this work.
- Partial literature lacks DOI number. Please standardize the references
The English could be improved to more clearly express the research. The language used should be more objective and logical.
Author Response

(The authors gave the same response as above.)

Reviewer 3 Report
Comments and Suggestions for Authors
This is a very interesting topic about kiwifruit. The author proposed a theoretical model and applied it to the purchasing attitude of kiwifruit, which has good practical value. Overall, the article is relatively long, and some tables and data can be included in the attachment. Some revisions are needed as follows.
- In the part of Introduction (line 33~34), please list the reasons for the high yield of kiwifruit in these regions.
- Please provide a detailed description of the types of kiwifruit. In addition to color, there are also nutritional value, taste, and so on.
- In Research Design, please further elaborate on it in different sections. For example, models, analysis methods, data processing.
- Screenshots should not be used for Tables. Please modify them according to the journal format.
- In Table 2.1, the texts in the Table cannot be seen clearly, please adjust them.
- In Table 10, what are the units of each data item?
- The research object of the article is kiwifruit, but no pictures of kiwifruit were seen. Please provide pictures of different types of kiwifruit for better reading.
- Please revise Concluding Remarks to the Conclusions.
Author Response

(The authors gave the same response as above.)

Round 2
Reviewer 1 Report
Comments and Suggestions for Authors
The manuscript has been sufficiently improved using the reviewer comments.
Authors need to adequately format their references and include DOI at proof reading stage.
Author Response
Many thanks to the anonymous reviewer. King regards
Reviewer 2 Report
Comments and Suggestions for Authors
While you have addressed several initial concerns, some issues remain unresolved and require immediate attention before publication can be considered.
- The reference format has not been optimized in the latest revised version. The response mentioned that it will be unified in the final revised version. Please further confirm.
- The reason for the lower adjusted R ² value still lacks further theoretical support. Although I acknowledge your explanation of the common low explanatory power in social sciences, this claim requires more literature to support it.
- The language and logic of the article require further refinement prior to its final submission.
Author Response
Thanks. Please refer to the attached Word

Reviewer 3 Report
Comments and Suggestions for Authors
The manuscript has been sufficiently improved.
Author Response

(The authors gave the same response as above.)
